# Nationwide Surveillance of Antifungal Resistance of *Candida* Bloodstream Isolates in South Korean Hospitals: Two Year Report from Kor-GLASS

**DOI:** 10.3390/jof8100996

**Published:** 2022-09-22

**Authors:** Eun Jeong Won, Min Ji Choi, Seok Hoon Jeong, Dokyun Kim, Kyeong Seob Shin, Jeong Hwan Shin, Young Ree Kim, Hyun Soo Kim, Young Ah Kim, Young Uh, Namhee Ryoo, Jeong Su Park, Kyoung Un Park, Seung A. Byun, Ga Yeong Lee, Soo Hyun Kim, Jong Hee Shin

**Affiliations:** 1Department of Laboratory Medicine, Chonnam National University Medical School, Gwangju 61469, Korea; 2Department of Laboratory Medicine, Research Institute of Bacterial Resistance, Yonsei University College of Medicine, Seoul 03722, Korea; 3Department of Laboratory Medicine, Chungbuk National University College of Medicine, Cheongju 28644, Korea; 4Department of Laboratory Medicine, Paik Institute for Clinical Research, Inje University College of Medicine, Busan 47392, Korea; 5Department of Laboratory Medicine, Jeju National University Medical School, Jeju 63243, Korea; 6Department of Laboratory Medicine, Hallym University Dongtan Sacred Heart Hospital, Hallym University College of Medicine, Hwaseong 18450, Korea; 7Department of Laboratory Medicine, National Health Insurance Service Ilsan Hospital, Goyang 10444, Korea; 8Department of Laboratory Medicine, Yonsei University Wonju College of Medicine, Wonju 26426, Korea; 9Department of Laboratory Medicine, Keimyung University School of Medicine, Daegu 42601, Korea; 10Department of Laboratory Medicine, Seoul National University Bundang Hospital, Seoul National University College of Medicine, Seoul 03080, Korea

**Keywords:** candidemia, *Candida* species, antifungal surveillance, GLASS

## Abstract

We incorporated nationwide *Candida* antifungal surveillance into the Korea Global Antimicrobial Resistance Surveillance System (Kor-GLASS) for bacterial pathogens. We prospectively collected and analyzed complete non-duplicate blood isolates and information from nine sentinel hospitals during 2020–2021, based on GLASS early implementation protocol for the inclusion of *Candida* species. *Candida* species ranked fourth among 10,758 target blood pathogens and second among 4050 hospital-origin blood pathogens. Among 766 *Candida* blood isolates, 87.6% were of hospital origin, and 41.3% occurred in intensive care unit patients. Adults > 60 years of age accounted for 75.7% of cases. Based on species-specific clinical breakpoints, non-susceptibility to fluconazole, voriconazole, caspofungin, micafungin, and anidulafungin was found in 21.1% (154/729), 4.0% (24/596), 0.1% (1/741), 0.0% (0/741), and 0.1% (1/741) of the isolates, respectively. Fluconazole resistance was determined in 0% (0/348), 2.2% (3/135, 1 Erg11 mutant), 5.3% (7/133, 6 Pdr1 mutants), and 5.6% (6/108, 4 *Erg11* and 1 *Cdr1* mutants) of *C. albicans, C. tropicalis, C. glabrata*, and *C. parapsilosis* isolates, respectively. An echinocandin-resistant *C. glabrata* isolate harbored an F659Y mutation in Fks2p. The inclusion of *Candida* species in the Kor-GLASS system generated well-curated surveillance data and may encourage global *Candida* surveillance efforts using a harmonized GLASS system.

## 1. Introduction

Antimicrobial resistance (AMR) is a leading cause of human death around the world [1]. In 2015, the World Health Organization (WHO) launched the Global Antimicrobial Resistance Surveillance and Use System (GLASS), the first global collaborative effort to standardize AMR surveillance [2]. GLASS enabled standardized AMR surveillance worldwide, which combined patient, laboratory, and epidemiological surveillance data to provide a picture of the extent and impact of AMR on populations. Following the principles of GLASS, the Centers for Disease Control and Prevention of South Korea (KCDC) established a customized AMR surveillance system for South Korea (Kor-GLASS) in 2017. This system received good reviews internationally as a desirable example of AMR surveillance [3,4]. In its early implementation phase (2015–2019), the aim of GLASS was global monitoring of AMR in common bacteria that cause infections in humans. However, recognizing the growing threat of resistant fungal infections, GLASS catalyzed a collaborative global effort to compile available data on antifungal-resistant infections and developed an early implementation protocol for the inclusion of *Candida* species in 2019 [5].

*Candida* bloodstream infections (BSIs) are the most common nosocomial fungal infections and are associated with high rates of mortality [6,7,8]. Nationwide surveillance of the antifungal resistance of *Candida* BSI isolates is important because it provides an overview of the emerging resistance of *Candida* species [5,7,9,10,11]. Moreover, knowledge of the rank order of occurrence and resistance profiles of various *Candida* species causing BSIs is important to establish empirical treatment protocols [6,10,12]. However, unlike bacteria, accurate identification and antifungal susceptibility testing of *Candida* species remain a major challenge, as many laboratories worldwide lack this capability [5]. Few countries have effective surveillance systems for fungal diseases and are thus ill-equipped to address the threat of antifungal-resistant fungi at the global level [5]. In response, the WHO developed the early implementation protocol to help countries create or strengthen their national fungal AMR surveillance systems [5]. We have incorporated a national AMR surveillance system for invasive *Candida* into the existing Kor-GLASS bacterial surveillance system., We also assessed antifungal resistance mechanisms for common *Candida* species in this *Candida* surveillance to understand the extent of resistance determinants in South Korean clinical settings, in line with the Kor-GLASS bacterial surveillance system [3]. In the following, we report the data obtained during the first 2 years (2020–2021) of the operation of the surveillance system for *Candida* species, which is based on the GLASS early implementation protocol.

## 2. Materials and Methods

### 2.1. Collection of Candida Blood Isolates

Along with Kor-GLASS bacterial pathogens, *Candida* blood isolates were prospectively collected from all patients with *Candida* BSIs at nine sentinel hospitals from January 2020 to December 2021. All hospitals were general hospitals that even treated pediatric patients, with a total of 7551 beds (655–1092 beds per hospital) and located in nine districts throughout South Korea (Appendix A). At each hospital, the first *Candida-positive* blood isolate per patient per species was collected. All isolates of *Candida* species in the MycoBank Database that had a *Candida* anamorph name or were previously called *Candida* were included [5]. The collected blood isolates were transferred to the analysis center (Chonnam National University Hospital) twice a month through a cold chain delivery system. For long-term storage, subsamples of each isolate were independently kept at three separate sites: one in the sentinel hospital, two in the analysis center, and two in the national reference laboratory [3].

### 2.2. Collection of Clinical Data from Sentinel Hospitals

Demographic data (age and sex), infection origin [hospital origin (HO) or community origin (CO)], and admission type (intensive care unit [ICU] or ward) were recorded at each sentinel hospital for all patients for whom blood cultures were performed during the study period [3,4,13,14]. An HO infection as indicated by a blood specimen taken from an inpatient hospitalized for 2 days, including the hospitalization days in another healthcare facility before transfer. A CO infection was indicated by a blood specimen taken from an out- or inpatient hospitalized for <2 days. All other considerations for Kor-GLASS bacterial pathogens were applied when setting up the national AMR surveillance system for *Candida* species [3,4]. The incidence of candidemia was calculated based on the number of cases of candidemia per 10,000 patient days (PD) at each hospital [15,16]. This study was approved by the Institutional Review Board (IRB) of Chonnam National University Hospital (CNUH-2020-080) and all sentinel hospitals. The IRBs of nine sentinel hospitals waived the requirement for informed consent because of the observational nature of the study and the very low risk of breaches of participant privacy.

### 2.3. Species Identification and Antifungal Susceptibility Testing

All collected isolates were re-identified using matrix-assisted laser desorption ionization time-of-flight (MALDI-TOF) mass spectrometry (Bruker Biotyper, Bruker Daltonics GmbH, Bremen, Germany) at the analysis center. Isolates with discrepant identification between the collection and the analysis centers, and isolates of uncommon species, were further analyzed by sequencing the D1/D2 domains of the 26S rRNA gene [7,11]. In vitro antifungal susceptibility testing of fluconazole, voriconazole, posaconazole, itraconazole, amphotericin B, caspofungin, micafungin, anidulafungin, and 5-flucytosine (5-FC) was performed using the Sensititre Yeast One (SYO) system (Thermo Scientific, Cleveland, OH, USA). Two reference strains, *Candida parapsilosis* ATCC 22019 and *Candida krusei* ATCC 6258 were included in each antifungal susceptibility test as quality control isolates. The interpretative guideline in the Clinical and Laboratory Standards Institute (CLSI) document M60 ED1 was used to classify isolates according to species-specific clinical breakpoints (CBPs) [17,18]. For external quality control of the species identification and antifungal susceptibility test results, 5% of the results obtained by the analysis center were reevaluated by comparison with those conducted in an independent quality control center. The analysis center was certified every 3 months by the external quality assurance program of the quality control center [13,14].

### 2.4. Molecular Mechanisms of Antifungal Resistance

The analysis center performed an advanced characterization of the isolates to assess the molecular mechanisms of antifungal resistance. The *ERG11* gene was sequenced in all fluconazole-resistant or susceptible dose-dependent (fluconazole minimum inhibitory concentration [MIC] ≥ 4 mg/L) isolates of *C. albicans, C. parapsilosis* and *C. tropicalis* [7,11,19]. The sequence of the *CDR1* gene was analyzed in fluconazole-resistant (fluconazole MIC ≥ 8 mg/L) *C. parapsilosis* isolates without mutations in *ERG11* [19]. *PDR1* was sequenced in all fluconazole-resistant (fluconazole MIC ≥ 64 mg/L) isolates of *C. glabrata* [11]. Echinocandin resistance was confirmed for all common *Candida* isolates with echinocandin MIC values higher than the CLSI CBPs (either intermediate [I] or resistant) through DNA sequence analysis of *FKS1* (all *Candida* species) and *FKS2*
*(C. glabrata*) [20].

### 2.5. Statistical Analysis

Fisher’s exact test or the chi-squared test was used to compare categorical variables, and a t-test was used to compare quantitative variables. All statistical analyses were performed using the Diagnostic Test Evaluation Calculator (MedCalc, Ostend, Belgium) and GraphPad Prism software (version 9.3.1; GraphPad Software Inc., San Diego, CA, USA). A *p*-value < 0.05 was considered to indicate statistical significance.

## 3. Results

### 3.1. Ranking of Candida Species among the Target Blood Pathogens

Among the 10,758 target blood pathogens obtained from Kor-GLASS (2020–2021), *Escherichia coli* (n = 4374, 40.7%) was the most common, followed by *Klebsiella pneumonia* (n = 1827, 17.0%), *Staphylococcus aureus* (n = 1501, 14.0%), *Candida* species (n = 766, 7.1%), *Enterococcus faecium* (n = 764, 7.1%), *Acinetobacter* species (n = 494, 4.6%), *Pseudomonas aeruginosa* (n = 454, 4.2%), *Enterococcus faecalis* (n = 448, 4.2%), *Streptococcus pneumoniae* (n = 82, 0.8%), and *Salmonella* species (n = 48, 0.4%) (Figure 1). Among the 4048 (37.6%) target blood pathogens of hospital origin, *E. coli* (n = 780) was the most common, followed by *Candida* species (n = 671), *S. aureus* (n = 657), *E. faecium* (n = 583), *K. pneumonia* (n = 498), *Acinetobacter* species (n = 386), *P. aeruginosa* (n = 241), and *E. faecalis* (n = 232). Among the 1983 BSI pathogens from ICU patients, *E. coli* (n = 362) was the most common, followed by *Candida* species (n = 316), *S. aureus* (n = 305), *K. pneumonia* (n = 265), *Acinetobacter* species (n = 252), *E. faecium* (n = 245), *P. aeruginosa* (n = 119), and *E. faecalis* (n = 103).

### 3.2. Species Distribution of Candida BSI Isolates

The species distributions of 766 non-duplicate bloodstream isolates of 15 *Candida* species are listed in Table 1. *C*. *albicans* was the most common of the *Candida* species identified, accounting for 45.4% of all cases, followed by *C. tropicalis* (17.6%), *C. glabrata* (17.4%), and *C. parapsilosis* (14.1%). The order of non-*albicans Candida* species (NAC) differed by year, but *C. tropicalis* and *C. glabrata* were the most common NAC in 2020 and 2021, respectively. Overall, the average incidence of candidemia was 1.58 cases per 10,000 PD, and the incidence of candidemia differed among the hospitals (1.06–2.22 cases per 10,000 PD). Species identification discrepancy between the collection center and the analysis center was noticed in nine (1.2%) isolates. The discrepancy rates varied according to the collection centers (0.0–4.4%). Two isolates of *C. albicans* were misidentified as *C. glabrata* and *C. tropicalis*, and two isolates of *C. parapsilosis* were misidentified as *C. albicans* and *C. tropicalis*. Four isolates (one isolate each of *C. glabrata*, *C. guilliermondii, C. dubliniensis,* and *C. ciferrii*) were misidentified as *C. albicans.* One isolate of *C. utilis* was misidentified as *C. fabianii.*

### 3.3. Clinical Characteristics of Patients with Candidemia

Of the 766 candidemia cases, patients aged >60 years accounted for 75.7% (580 cases), and pediatric patients (aged 1–19 years) accounted for 1.9% (15 cases) (Figure 2). *C. glabrata* BSI was more frequent in the elderly (>70 years) than BSIs caused by all other *Candida* species (60.9% vs. 50.6%, *p* = 0.029). In total, there were 450 (58.7%) cases in men, 671 (87.6%) were classified as HO, and 316 (41.3%) occurred in the ICU (Figure 3A–C). *C. parapsilosis* BSIs were more frequent than BSIs caused by other *Candida* species in men (70.4% vs. 56.8%, *p =* 0.008), and *C. glabrata* BSIs were more frequently associated with CO candidemia than were BSIs caused by other *Candida* species (18.0% vs. 11.2%, *p* = 0.034)**.** Outpatients accounted for 5.9% of all candidemia cases and inpatients for 94.1%.

### 3.4. Antifungal Resistance of Candida Bloodstream Isolates

Table 2 shows the antifungal susceptibility of 741 BSI isolates of six common *Candida* species to antifungal agents for which CLSI CBPs were available. Based on species-specific CBPs, non-susceptibility to fluconazole, voriconazole, caspofungin, micafungin, and anidulafungin was determined in 21.1% (154/729), 4.0% (24/596), 0.1% (1/741), 0.0% (0/741), and 0.1% (1/741) of the isolates, respectively. The rate of fluconazole resistance was 0% (0/348) in *C. albicans*, but was higher in NAC, particularly *C. parapsilosis* (5.6%, 6/108), *C. glabrata* (5.3%, 7/133), and *C. tropicalis* (2.2%, 3/135). All isolates were susceptible to echinocandins, except for a *C. glabrata* isolate that was resistant to caspofungin (MIC, 0.5 µg/mL) and had intermediate resistance to anidulafungin (MIC, 0.25 µg/mL). The rate of fluconazole resistance was higher in isolates from ICU patients (14/301, 4.7%) than ward patients (7/428, 1.6%) (*p =* 0.017) (Figure 3D). The rate of fluconazole resistance was significantly higher in the ICU than ward for *C. glabrata* (9.5% vs. 1.4%, *p =* 0.037), but not for the other species. Table 3 lists the MIC range, MIC_50_ and MIC_90_ values of antifungal agents for which CBPs were not available for six common *Candida* species. There were no significant differences in the MIC_50_ and MIC_90_ values of these species between 2020 and 2021. The MIC distributions of nine antifungal agents for 25 isolates of nine uncommon species of *Candida* are shown in Table 4. Although CBPs were not available, four isolates had elevated fluconazole MIC values (≥8 mg/L), including one isolate each of *C. auris*, *C. fabianii*, *C. orthopsilosis*, and *C. haemulonii.* One isolate of *C. haemulonii* had elevated MIC values for amphotericin B (8 mg/L), fluconazole (64 mg/L), voriconazole (8 mg/L), and posaconazole (8 mg/L).

### 3.5. Molecular Mechanisms of Antifungal Resistance

Table 5 summarizes the antifungal resistance mechanisms of 24 *Candida* bloodstream isolates. All 23 fluconazole non-susceptible *Candida* isolates were from HO infections. Of the seven fluconazole-resistant isolates of *C. glabrata* from six hospitals, six harbored Pdr1p amino acid substitutions (E259G, C469R, F559S, T745A, or S942F). Of the eight fluconazole non-susceptible *C. parapsilosis* isolates, five exhibited either a Y132F mutation (four isolates from one hospital) or K143R mutation (one isolate from another hospital) in Erg11p, and one isolate from another hospital harbored an N1132D mutation in Cdr1p. Of the eight fluconazole non-susceptible isolates of *C. tropicalis* from four hospitals, two exhibited a Y132F or Y257H mutation in Erg11p. A caspofungin-resistant and anidulafungin intermediate *C. glabrata* isolate harbored an F659Y mutation in *FKS2*.

## 4. Discussion

To the best of our knowledge, this is the first study to incorporate a national surveillance system for *Candida* BSIs into the GLASS bacterial surveillance system. The results showed that a Kor-GLASS-like surveillance system including isolate collection and centralized analysis components can provide reliable data for both *Candida* and bacterial blood pathogens. During the two-year study period, 10,758 BSI cases were caused by 10 Kor-GLASS target pathogens, and 4048 (37.6%) BSIs were classified as HO. *Candida* species ranked fourth among all target blood pathogens, and second among all HO blood pathogens, thus highlighting the important casual role of *Candida* species in nosocomial BSIs in South Korea. Surveillance at nine sentinel hospitals showed that 87.6% of candidemia cases were of HO, similar to the findings of an Australian study [21]. The average incidence of candidemia was 1.58 cases per 10,000 PD, but the incidence differed among hospitals (range: 1.06–2.22), as also reported in a previous Asian study [22].

In earlier multicenter studies conducted in South Korea, *C.*
*parapsilosis* was the most common NAC isolated from patients with candidemia [16,23,24]. However, we found that, for the period from 2020–2021, *C. glabrata* and *C. tropicalis* were the most common NACs causing candidemia, consistent with recent reports on the changes in species distributions of candidemia in South Korea [8,25]. In studies from the USA, northern Europe, and Australia, *C. glabrata*, which is less susceptible to antifungal drugs, was the second most common cause of candidemia [10,21,26], while in tropical and Asian countries, such as The Philippines and Thailand, *C.*
*tropicalis* was the most prevalent NAC [22,27]. In the present study, most candidemia cases occurred in patients aged >60 years with *C. glabrata* BSIs being more frequent in the elderly (>70 years) and *C. parapsilosis* candidemia being more frequent in males, in line with our recent study [8]. The reasons for the changing epidemiology of NAC BSIs in South Korea are unclear, but the use of antifungal agents, infection control practices, and types of at-risk hospitalized patients enrolled may be factors [5,8,10,26]. Given that each *Candida* species is unique in terms of virulence potential, antifungal susceptibility, and clinical characteristics, understanding the changing epidemiology is important for the proper management of candidemia [8,28]. In South Korea, fluconazole or amphotericin B were mainly used for the treatment of candidemia until the approval of echinocandins as primary treatment for severe candidiasis by the National Health Insurance Service (NHIS) in 2014. Following the approval of echinocandin use by the NHIS, the use of echinocandins for candidemia treatment has increased after 2014, and attention is required because of echinocandin resistance emergence [29,30].

Sensititre Yeast One was used for surveillance in this study, as it has been widely adopted by clinical microbiology laboratories for antifungal susceptibility testing and shows good concordance with the CLSI reference method for *Candida* susceptibility testing [31]. In addition, we examined several molecular mechanisms of antifungal resistance to better understand AMR epidemiology. While echinocandin resistance remains rare in Korea for six common *Candida* species (<0.5%), we observed non-susceptibility to fluconazole in about 20% of these species, in addition to frequent azole resistance among BSI isolates of three common NACs. The singular mechanism of acquired azole resistance identified in clinical isolates of *C. glabrata* is mutation of the transcription factor pleiotropic drug-resistance (*PDR1*), which leads to overexpression of the drug-efflux transporter genes *CgCDR1, CgCDR2*, and *CgSNQ2* [11,31]. Our recent Korean study showed that 98.5% of fluconazole-resistant BSI isolates of *C. glabrata* and 0.9% of fluconazole susceptible dose-dependent BSI isolates of *C. glabrata* harbored an additional one or two Pdr1p AAS after exclusion of five genotype-specific AAS. In addition, the results highlight the high mortality rate of patients infected with fluconazole-resistant *C. glabrata* BSI isolates harboring Pdr1p mutations [11]. By contrast, multiple mechanisms of azole resistance, such as *ERG11* mutations and overexpression of efflux pumps, have been reported for *C. albicans, C. parapsilosis* and *C.*
*tropicalis* [31]; however, Y132F in *ERG11* and N1132D in *CDR1* were the major mechanisms of fluconazole resistance in *C.*
*parapsilosis* isolates from Korean hospitals [7,19]. Therefore, the *PDR1* gene was sequenced for all fluconazole-resistant isolates of *C. glabrata* while *ERG11* and *CDR1* genes were sequenced for all fluconazole-non-susceptible isolates of *C. parapsilosis*, and *ERG11* gene was sequenced for all fluconazole-non-susceptible isolates of *C. tropicalis*, although other mechanisms might contribute.

Six of the seven fluconazole-resistant isolates of *C. glabrata* harbored diverse Pdr1p amino acid substitutions, which suggests that *PDR1* mutation is the main cause of azole resistance in *C. glabrata,* in line with previous studies [11,32]. An azole-resistant *C. parapsilosis* isolate harbored a N1132D substitution in *CDR1,* as in another recent report [19]. Notably, all four fluconazole-non-susceptible *C. parapsilosis* isolates harboring a Y132F substitution in Erg11p were collected at the same hospital, indicating possible clonal transmission of these isolates [7]. One *C. glabrata* isolate was classified as caspofungin-resistant (MIC, 0.5 mg/L), with intermediate resistance to anidulafungin (MIC, 0.25 mg/L) but susceptibility to micafungin (MIC, 0.06 mg/L). In this isolate, echinocandin resistance was determined by DNA sequence analysis of the *FKS* genes. The isolate harbored an F659Y mutation in Fks2p, which confers resistance to all echinocandins [17].

An increasing incidence of candidemia in ICUs has been reported in many parts of the world [15,33,34,35]. In a previous study based on Korean National Healthcare-Associated Infections Surveillance System, which included patients older than 15 years who developed candidemia during a stay of >2 days in the ICU, *Candida* species were the most frequently identified blood pathogens from 2013 to 2017 [15]. In the present study, 41.3% of candidemia cases occurred in ICU patients, and *Candida* species were the second most common BSI pathogen recovered from these patients (after *E. coli*). The rate of fluconazole resistance was significantly higher in ICU patients (4.7%) than ward patients (1.6%), as was the fluconazole resistance of *C. glabrata* (9.5% vs. 1.4%). Our recent study showed that, among patients with candidemia, mortality rates were approximately twofold higher in ICU than ward patients, and that fluconazole resistance was a predictor of *C.-glabrata*-associated mortality [8]. Taken together, these results highlight the importance of candidemia control in ICUs, including improved prevention and treatment strategies such as optimal use of antifungal treatments and less unnecessary use of catheters.

Accurate species identification is crucial considering the increasing rate of antifungal resistance among uncommon *Candida* species, such as *Candida*
*auris* [36]. Our surveillance system used MALDI-TOF MS, supplemented by sequence-based identification of *Candida* species at an analysis center. In our study, discrepant identification results between the collection and analysis centers were seen for only 1.2% of the isolates (from five common and four uncommon *Candida* species). Given that all nine hospitals participating in Kor-GLASS used the MALDI-TOF MS system for the identification of yeast isolates, the misidentification of five isolates of common *Candida* species might be worthy of note. We found that a collection center using the MALDI-TOF Biotyper, which did not use FA extraction routinely, exhibited a relatively high misidentification rate (4.4%), even in common *Candida* species. For the analysis using the MALDI-TOF Biotyper, in-tube formic acid/acetonitrile (FA/ACN) extraction is recommended prior to the analysis; however, the use of a simple FA extraction method is preferable in order to facilitate the routine use in clinical microbiology laboratories [37]. Therefore, it should be considered that although MALDI-TOF system allows reliable and accurate identification of the clinical isolates of yeast species, their performance can be differed according to the protein extraction protocol (i.e., use of formic acid or not) or MALDI-TOF MS system databases used at the hospitals [37,38].

For some BSI isolates of rare *Candida* species for which CBPs were unavailable, including *C. auris, C. fabianii,* and *C. orthopsilosis*, resistance to fluconazole was higher than for other species (fluconazole MIC > 8 mg/L). In addition, a *C. haemulonii* isolate was determined to be multidrug-resistant (i.e., to both azole and amphotericin B), as reported previously [39]. These results suggest that the distribution of azole- and echinocandin-resistant NAC isolates should be continuously monitored at the national level, and the resistance mechanisms determined.

Several nationwide surveillance studies of the antifungal susceptibility of *Candida* species causing BSIs in South Korea have been published since 2007 [7,8,11,16,23,24,40,41]. A total of 20 university hospitals in South Korea participated in those studies, but the participating hospitals differed among years. In addition, in some of the studies, more than half of the *Candida* BSI isolates were from the two largest hospitals (>2000 beds), both of which are in Seoul, which is also where most of the antifungal-resistant isolates were found [7,11]. Also, in this study, hospital H, located at Jeju-island, off the peninsula’s southern tip, showed a bit of a different pattern in the epidemiology of candidemia, in which *C. albicans* did not exceed NAC. It can be partly explained by their geographical and ecological conditions that may have influenced the epidemiology of candidiasis. Overall, the incidence rate and AMR of *Candida* BSIs vary greatly among institutions, with factors such as the number of beds, patients, and PD in the ICU or hemato-oncology wards likely playing important roles. Although whether the participating hospitals are representative is always debatable in national candida AMR surveillance studies, this concern can be partly overcome by incorporating a national surveillance system for BSI into the GLASS system, which permits continuous monitoring of target blood pathogens and is free from collection bias and isolate duplication. The Kor-GLASS system was designed to collect and analyze complete, non-duplicate clinical isolates and information from sentinel hospitals located in various districts throughout South Korea (each with a capacity of 600–1100 beds) caring for both inpatients and outpatients. The surveillance data demonstrated both the changing epidemiology of *Candida* species causing BSIs during 2020–2021, and the continued emergence of azole and echinocandin resistance (and thus of new resistance mechanisms) among BSI isolates of NAC. Continuous monitoring is therefore warranted.

## 5. Conclusions

We incorporated AMR surveillance for invasive *Candida* into preexisting GLASS bacterial surveillance system, and herein, we describe results from the two year (2020–2021) of operation of Kor-GLASS *Candida* surveillance based on early implementation protocol. The well-curated surveillance data highlight the significance of candidemia both as the second-most common cause of BSIs of HO and the second-most common pathogen in ICU patients in South Korea. Our study showed that the inclusion of *Candida* species in the Kor-GLASS system allowed representative and accurate monitoring of the antifungal resistance of BSI-causing candida on a national scale. As such, it will support further global efforts aimed at candidemia AMR surveillance.

## Figures and Tables

**Figure 1 jof-08-00996-f001:**
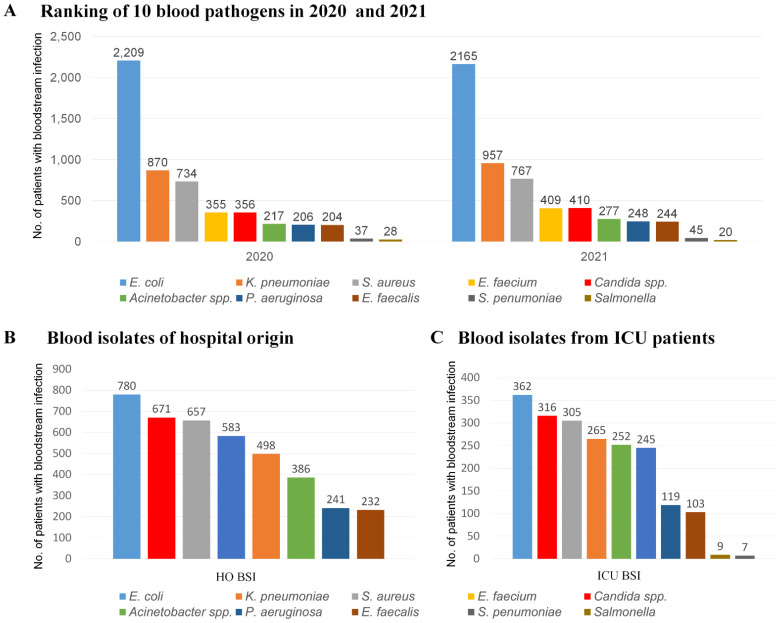
Number of patients with bloodstream infections (BSIs) for each target pathogen collected via the Kor-GLASS surveillance system (2010–2021). (**A**) Total number of BSIs caused by each target pathogen. (**B**) Number of hospital-origin BSIs caused by each pathogen. (**C**) Number of BSIs caused by each pathogen in ICU patients.

**Figure 2 jof-08-00996-f002:**
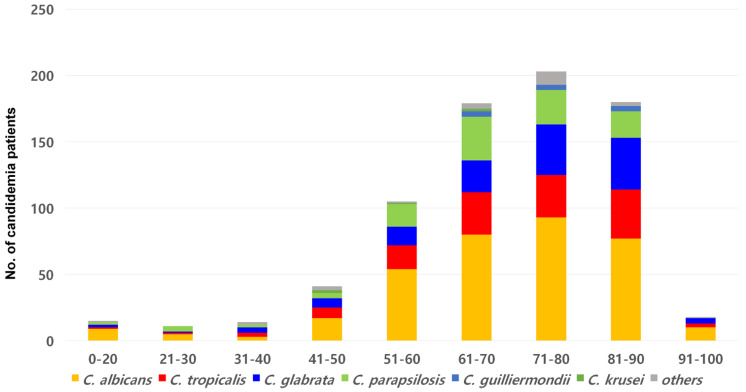
Overall distributions of the *Candida* species causing BSIs by candidemia patient age group.

**Figure 3 jof-08-00996-f003:**
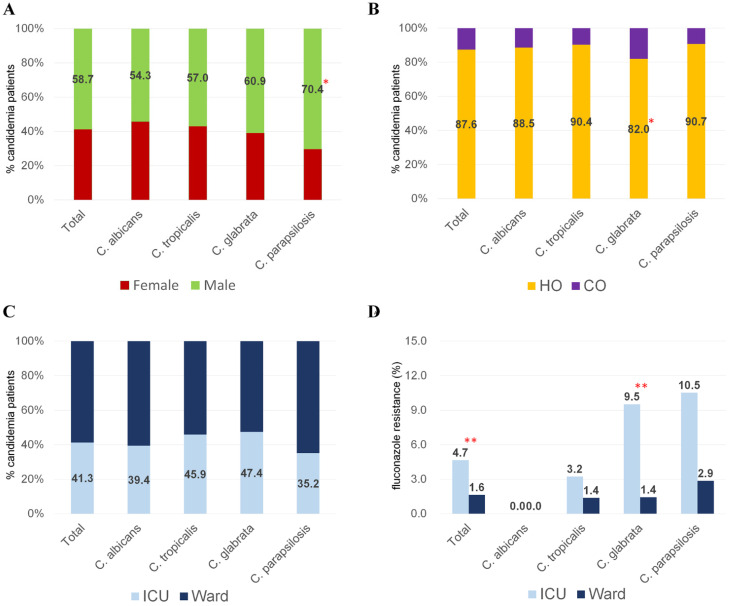
Clinical information of cases of candidemia caused by different *Candida* species: sex (**A**), origin (hospital origin [HO] or community origin [CO]) (**B**) and location (ICU vs. ward) (**C**). In (**A**–**C**), the results are based on the 766 patients enrolled in this study. Rates of fluconazole resistance of *Candida* isolates recovered from the ICU and wards (**D**): the 729 isolates included *C. albicans* (n = 348), *C. tropicalis* (n = 135), *C. glabrata* (n = 133), *C. parapsilosis* (n = 108), and *C. krusei* (n = 5), for which clinical breakpoints were available in the CLSI M60. * *p* < 0.05 between a *Candida* species and all other *Candida* species within a given category (i.e., number of patients, sex [female vs. male], origin [HO vs. CO], or location [ICU vs. ward]). ** *p* < 0.05 between ICU and ward patients in the rate of fluconazole resistance according to the *Candida* species.

**Table 1 jof-08-00996-t001:** The incidence of candidemia collected from Kor-GLASS surveillance system during the two-year periods.

*Candida* Species	No. of Candidemia Episodes at Each Hospital									No (%) in Each Year:		
	A	B	C	D	E	F	G	H	I	2020	2021	Total
*C. albicans*	55	25	40	36	48	43	28	10	63	172 (48.3)	176 (42.9)	348 (45.4)
*C. tropicalis*	16	15	10	13	21	12	13	5	30	61 (17.1)	74 (18)	135 (17.6)
*C. glabrata*	17	9	27	7	15	16	16	11	15	57 (16)	76 (18.5)	133 (17.4)
*C. parapsilosis*	18	6	15	10	12	11	11	10	15	46 (12.9)	62 (15.1)	108 (14.1)
*C. guilliermondii*	1	0	0	2	0	1	4	1	3	10 (2.8)	2 (0.5)	12 (1.6)
*C. lusitaniae*	3	0	2	0	2	1	0	0	1	4 (1.1)	5 (1.2)	9 (1.2)
*C. krusei*	0	0	2	1	1	0	0	1	0	2 (0.6)	3 (0.7)	5 (0.7)
*C. fabianii*	0	0	0	0	1	2	1	0	0	1 (0.3)	3 (0.7)	4 (0.5)
*C. dubliniensis*	0	0	1	0	1	0	0	0	1	1 (0.3)	2 (0.5)	3 (0.4)
*C. metapsilosis*	1	0	0	0	2	0	0	0	0	0 (0)	3 (0.7)	3 (0.4)
*C. orthopsilosis*	1	0	1	0	0	0	0	0	0	0 (0)	2 (0.5)	2 (0.3)
*C. pelliculosa*	0	0	0	0	0	0	0	0	1	0 (0)	1 (0.2)	1 (0.1)
*C. bracarensis*	1	0	0	0	0	0	0	0	0	1 (0.3)	0 (0)	1 (0.1)
*C. haemulonii*	0	0	0	0	1	0	0	0	0	1 (0.3)	0 (0)	1 (0.1)
*C. auris*	0	0	0	0	1	0	0	0	0	0 (0)	1 (0.2)	1 (0.1)
Total No.	113	55	98	69	105	86	73	38	129	356 (100)	410 (100)	766 (100)
Incidence/10,000 patient-days												
	1.92	1.07	2.22	1.15	1.76	1.69	1.48	1.06	1.76	1.45	1.72	1.58

**Table 2 jof-08-00996-t002:** Antifungal susceptibilities of 741 isolates of six common *Candida* species which clinical breakpoints are available in CLSI M60.

Antifungal Agent * /Species		2020			2021			Total			
		No	%R	%SDD/I	No	%R	%SDD/I	No	%R	%SDD/I	%NS
FLU	*C*. *albicans*	172	0.0	0.0	176	0.0	0.0	348	0.0	0.0	0.0
	*C*. *tropicalis*	61	1.6	1.6	74	2.7	5.4	135	2.2	3.7	5.9
	*C*. *glabrata*	57	5.3	94.7	76	5.3	94.7	133	5.3	94.7	100.0
	*C*. *parapsilosis*	46	4.3	0.0	62	6.5	3.2	108	5.6	1.9	7.4
	*C*. *krusei*	2	100.0	0.0	3	100.0	0.0	5	100.0	0.0	100.0
	Total	338	2.4	16.3	391	3.3	19.9	729	2.9	18.2	21.1
VOR	*C*. *albicans*	172	0.0	0.0	176	0.0	0.0	348	0.0	0.0	0.0
	*C*. *tropicalis*	61	1.6	9.8	74	2.7	13.5	135	2.2	11.9	14.1
	*C*. *parapsilosis*	46	0.0	2.2	62	0.0	6.5	108	0.0	4.6	4.6
	*C*. *krusei*	2	0.0	0.0	3	0.0	0.0	5	0.0	0.0	0.0
	Total	281	0.4	2.5	315	0.6	4.4	596	0.5	3.5	4.0
CAS	*C*. *albicans*	172	0.0	0.0	176	0.0	0.0	348	0.0	0.0	0.0
	*C*. *tropicalis*	61	0.0	0.0	74	0.0	0.0	135	0.0	0.0	0.0
	*C*. *glabrata*	57	0.0	0.0	76	1.3	0.0	133	0.8	0.0	0.8
	*C*. *parapsilosis*	46	0.0	0.0	62	0.0	0.0	108	0.0	0.0	0.0
	*C*. *guilliermondii*	10	0.0	0.0	2	0.0	0.0	12	0.0	0.0	0.0
	*C*. *krusei*	2	0.0	0.0	3	0.0	0.0	5	0.0	0.0	0.0
	Total	348	0.0	0.0	393	0.3	0.0	741	0.1	0.0	0.1
MICA	*C*. *albicans*	172	0.0	0.0	176	0.0	0.0	348	0.0	0.0	0.0
	*C*. *tropicalis*	61	0.0	0.0	74	0.0	0.0	135	0.0	0.0	0.0
	*C*. *glabrata*	57	0.0	0.0	76	0.0	0.0	133	0.0	0.0	0.0
	*C*. *parapsilosis*	46	0.0	0.0	62	0.0	0.0	108	0.0	0.0	0.0
	*C*. *guilliermondii*	10	0.0	0.0	2	0.0	0.0	12	0.0	0.0	0.0
	*C*. *krusei*	2	0.0	0.0	3	0.0	0.0	5	0.0	0.0	0.0
	Total	348	0.0	0.0	393	0.0	0.0	741	0.0	0.0	0.0
ANI	*C*. *albicans*	172	0.0	0.0	176	0.0	0.0	348	0.0	0.0	0.0
	*C*. *tropicalis*	61	0.0	0.0	74	0.0	0.0	135	0.0	0.0	0.0
	*C*. *glabrata*	57	0	0	76	0.0	1.3	133	0.0	0.8	0.8
	*C*. *parapsilosis*	46	0.0	0.0	62	0.0	0.0	108	0.0	0.0	0.0
	*C*. *guilliermondii*	10	0.0	0.0	2	0.0	0.0	12	0.0	0.0	0.0
	*C*. *krusei*	2	0.0	0.0	3	0.0	0.0	5	0.0	0.0	0.0
	Total	348	0	0	393	0.0	0.3	741	0.0	0.1	0.1

Abbreviations: R, resistant; SDD, susceptible dose-dependent; I, intermediate; NS, non-susceptible; FLU, fluconazole; VOR, voriconazole; CAS, caspofungin; ANI, anidulafungin; MICA, micafungin. * Numbers (%) of isolates categorized by the Clinical and Laboratory Standards Institute (CLSI M60) species-specific breakpoints [17].

**Table 3 jof-08-00996-t003:** Antifungal susceptibilities of 741 isolates of six common *Candida* species which clinical breakpoints are not available in CLSI M60.

Antifungal Agent/ Species		MICs * (mg/L) in 2020				MICs * (mg/L) in 2021			
		No.	Range	MIC_50_	MIC_90_	No.	Range	MIC_50_	MIC_90_
FLU	*C*. *guilliermondii*	10	4–16	4	16	2	4–8	4	8
VOR	*C*. *glabrata*	57	0.03–4	0.5	1	76	0.12–4	0.5	1
	*C*. *guilliermondii*	10	0.06–0.25	0.25	0.25	2	0.06–0.25	0.25	0.25
POS	*C*. *albicans*	172	0.008–0.25	0.03	0.03	176	0.008–0.25	0.03	0.06
	*C*. *glabrata*	57	0.5–16	1	2	76	0.5–>8	1	2
	*C*. *tropicalis*	61	0.03–0.5	0.12	0.5	74	0.03–0.5	0.12	0.25
	*C*. *parapsilosis*	46	0.015–0.12	0.03	0.06	62	0.015–0.25	0.06	0.12
	*C*. *guilliermondii*	10	0.12–0.5	0.25	0.5	2	0.25–0.5	0.25	0.5
	*C*. *krusei*	2	0.12–0.25	0.12	0.25	3	0.25–0.5	0.5	0.5
ITC	*C. albicans*	172	0.008–0.12	0.03	0.12	176	0.015–0.25	0.06	0.12
	*C. glabrata*	57	0.06–>16	0.5	1	76	0.25–>16	1	1
	*C. tropicalis*	61	0.06–0.5	0.25	0.5	74	0.12–1	0.25	0.5
	*C. parapsilosis*	46	0.008–0.25	0.06	0.12	62	0.015–0.25	0.06	0.12
	*C. guilliermondii*	10	0.25–0.5	0.5	0.5	2	0.25–0.5	0.25	0.5
	*C. krusei*	2	0.12–0.25	0.12	0.25	3	0.25–0.5	0.25	0.5
AMB	*C*. *albicans*	172	0.12–1	0.5	1	176	0.12–1	0.5	1
	*C*. *glabrata*	57	0.5–2	1	1	76	0.25–1	1	1
	*C*. *tropicalis*	61	0.25–1	1	1	74	0.5–1	1	1
	*C*. *parapsilosis*	46	0.12–2	0.5	1	62	0.12–1	0.5	1
	*C*. *guilliermondii*	10	0.12–0.5	0.25	0.5	2	0.12–0.25	0.12	0.25
	*C*. *krusei*	2	1	1	1	3	0.5–1	1	1
5-FC	*C*. *albicans*	172	0.06–128	0.06	0.25	176	0.06–>64	0.06	0.25
	*C*. *glabrata*	57	0.06–2	0.06	0.06	76	0.06–2	0.06	0.06
	*C*. *tropicalis*	61	0.008–0.25	0.06	0.12	74	0.06–0.12	0.06	0.06
	*C*. *parapsilosis*	46	0.008–0.12	0.06	0.06	62	0.06–0.25	0.06	0.06
	*C*. *guilliermondii*	10	0.06	0.06	0.06	2	0.06	0.06	0.06
	*C*. *krusei*	2	8	8	8	3	8–16	8	16

* The MIC_50_ and MIC_90_, MICs at which 50% and 90% of isolates are inhibited, respectively. Abbreviations: MIC, minimum inhibitory concentration; FLU, fluconazole; VOR, voriconazole; POS, posaconazole; ITC, itraconazole; AMB, amphotericin B; 5-FC, flucytosine.

**Table 4 jof-08-00996-t004:** Distributions of MIC values to antifungal agents of uncommon *Candida* species which clinical breakpoints are unavailable in CLSI M60.

Antifungal Agent/Species	N of Isolates for Which the MIC (mg/L) Was:														
	0.01	0.02	0.03	0.06	0.12	0.25	0.5	1	2	4	8	16	32	64	≥128
FLU															
*C. lusitaniae*						1	6	2							
*C. fabianii*							2	1						1	
*C. dubliniensis*					1	2									
*C. metapsilosis*								2	1						
*C. orthopsilosis*									1				1		
*C. auris*															1
*C. bracarensis*									1						
*C. haemulonii*															1
*C. pelliculosa*									1						
VOR															
*C. lusitaniae*	3	6													
*C. fabianii*	1		2				1								
*C. dubliniensis*	2	1													
*C. metapsilosis*		1	2												
*C. orthopsilosis*					1			1							
*C. auris*								1							
*C. bracarensis*				1											
*C. haemulonii*											1				
*C. pelliculosa*				1											
POS															
*C. lusitaniae*			9												
*C. fabianii*				1		2	1								
*C. dubliniensis*		1	1	1											
*C. metapsilosis*		1	1	1											
*C. orthopsilosis*						2									
*C. auris*						1									
*C. bracarensis*						1									
*C. haemulonii*											1				
*C. pelliculosa*						1									
ITC															
*C. lusitaniae*				4	5										
*C. fabianii*			1		1	1	1								
*C. dubliniensis*			2	1											
*C. metapsilosis*		1		1	1										
*C. orthopsilosis*						2									
*C. auris*						1									
*C. bracarensis*						1									
*C. haemulonii*													1		
*C. pelliculosa*				1											
CAS															
*C. lusitaniae*					5	4									
*C. fabianii*			1	1	2										
*C. dubliniensis*			1	2											
*C. metapsilosis*				2		1									
*C. orthopsilosis*						1	1								
*C. auris*				1											
*C. bracarensis*				1											
*C. haemulonii*					1										
*C. pelliculosa*			1												
MICA															
*C. lusitaniae*				9											
*C. fabianii*			2	2											
*C. dubliniensis*		2	1												
*C. metapsilosis*					1		2								
*C. orthopsilosis*							2								
*C. auris*			1												
*C. bracarensis*		1													
*C. haemulonii*						1									
*C. pelliculosa*			1												
ANI															
*C. lusitaniae*					9										
*C. fabianii*			1		3										
*C. metapsilosis*				1	1		1								
*C. orthopsilosis*							2								
*C. auris*				1											
*C. bracarensis*		1													
*C. haemulonii*					1										
*C. pelliculosa*		1													
*C. dubliniensis*				1	2										
AMB															
*C. lusitaniae*						2	7								
*C. fabianii*						1	2	1							
*C. dubliniensis*						1	2								
*C. metapsilosis*						1	2								
*C. orthopsilosis*							2								
*C. auris*						1									
*C. bracarensis*							1								
*C. haemulonii*											1				
*C. pelliculosa*						1									
5-FC															
*C. lusitaniae*				9											
*C. fabianii*				4											
*C. dubliniensis*				3											
*C. metapsilosis*				3											
*C. orthopsilosis*				2											
*C. auris*					1										
*C. bracarensis*				1											
*C. haemulonii*				1											
*C. pelliculosa*											1				

Abbreviations: FLU, fluconazole; VOR, voriconazole; POS, posaconazole; ITC, itraconazole; CAS, caspofungin; ANI, anidulafungin; MICA, micafungin; AMB, amphotericin B; 5-FC, flucytosine.

**Table 5 jof-08-00996-t005:** Antifungal resistance mechanisms of 24 fluconazole or echinocandins resistant or non-susceptible *Candida* bloodstream isolates obtained from Kor-GLASS surveillance (2020–2021).

Year/Origin	Sex/Age/ICU	Species	MIC (µg/mL)		Amino Acid Substitutions				Genbank Accession No. *
			FLU/VOR/POS/ITC	CAS/ANI/MICA	Erg11p	Cdr1p	Pdr1p	Fks2p	
2020/HO	M/85/Ward	CG	64/1/2/2	0.06/0.03/0.015			E259G		OP125549
2020/HO	M/80/ICU	CG	256/4/>8/>16	0.12/0.03/0.015			C469R		OP125550
2020/HO	F/73/ICU	CG	128/2/>8/>16	0.12/0.03/0.015			F559S		OP125551
2021/HO	F/80/ICU	CG	128/2/2/2	0.12/0.06/0.03			T745A		OP125552
2021/HO	F/68/ICU	CG	64/1/2/1	0.06/0.015/0.015			E259G		OP125553
2021/HO	M/77/ICU	CG	128/2/2/2	0.06/0.03/0.015			None		OP125554
2021/HO	M/78/ICU	CG	128/4/>8/>16	0.12/0.03/0.015			S942F		OP125555
2020/HO	F/78/Ward	CP	8/0.06/0.015/0.03	0.25/1/1	Y132F				OP125556
2020/HO	M/48/ICU	CP	64/0.25/0.12/0.25	0.5/1/1	K143R				OP125557
2021/HO	F/77/ICU	CP	32/0.25/0.03/0.06	0.5/1/2	Y132F		-	-	OP125558
2021/HO	F/53/ICU	CP	32/0.25/0.03/0.06	0.25/1/1	Y132F				OP125559
2021/HO	F/67/ICU	CP	16/0.5/0.25/0.25	0.5/0.5/1	None	N1132D			OP125560 (*ERG11*), OP125572 (*CDR1*)
2021/HO	M/84/Ward	CP	8/0.25/0.25/0.25	0.5/1/1	None	None			OP125561 (*ERG11*), OP125573 (*CDR1*)
2021/HO	F/53/Ward	CP	4/0.12/0.25/0.25	0.5/1/2	None				OP125567
2021/HO	F/38/Ward	CP	4/0.12/0.03/0.06	0.25/1/2	Y132F				OP125566
2020/HO	F/78/ICU	CT	32/2/0.25/0.25	0.06/0.06/0.03	Y132F				OP125562
2020/HO	M/82/ICU	CT	4/0.25/0.5/0.5	0.03/0.015/0.03	None				OP125568
2021/HO	M/65/ICU	CT	8/1/1/1	0.06/0.06/0.03	None				OP125563
2021/HO	M/39/Ward	CT	8/1/0.25/0.25	0.06/0.06/0.03	None				OP125564
2021/HO	F/93/Ward	CT	4/0.5/0.25/0.25	0.03/0.06/0.015	None				OP125569
2021/HO	F/49/Ward	CT	4/0.5/0.5/0.5	0.03/0.06/0.03	Y257H				OP125565
2021/HO	F/54/ICU	CT	4/0.5/0.5/0.5	0.03/0.015/0.015	None				OP125570
2021/HO	M/65/ICU	CT	4/0.12/0.06/0.12	0.06/0.015/0.03	None				OP125571
2021/CO	F/66/Ward	CG	8/0.25/1/0.5	0.5/0.25/0.06				F659Y	OP125574

* Sequences were deposited in GenBank under accession numbers OP125549-OP125571, OP125574, respectively. Abbreviations: HO, hospital Origin; CO, community origin; ICU, intensive care unit; FLU, fluconazole; VOR, voriconazole; POS, posaconazole; ITC, itraconazole; CAS, caspofungin; ANI, anidulafungin; MICA, micafungin; CG, *C. glabrata*; CP, *C. parapsilosis;* CT, *C. tropicals;* none, no amino acid substitutions.

## Data Availability

All data generated or analyzed in this study are included in this published article, and the datasets are available from the corresponding author within the limits imposed by ethical and legal dispositions.

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
