# Peer review of "Nationwide Surveillance of Antifungal Resistance of Candida Bloodstream Isolates in South Korean Hospitals: Two Year Report from Kor-GLASS"

_jof, 2022, doi:10.3390/jof8100996_

Round 1

Reviewer 1 Report

Review of the article: Nationwide Surveillance of Antifungal Resistance of Candida Bloodstream Isolates in South Korean Hospitals: Two Year Report from Kor-GLASS

Manuscript ID: jof-1887584

In my opinion, the proposed manuscript is very interesting and generally well prepared. The number of Candida spp. strains tested is impressive. The process of collecting of the samples was performed in a fully professional manner. I do not have any doubts that presented results are interesting and important for researchers (but also medical doctors) who work in a similar research area. However, I have some critical remarks about the manuscript – the detailed comments are presented below. I would be grateful if the authors could address my comments and suggestions.

Detailed comments

Abstract – well prepared and informative, no critical comments.

Introduction – short, but well prepared.

Materials and methods

Figure 1 – it is not clear for me what do the authors mean by “Patient day” and why this “parameter was used”.

Lines 112 – 113 – as far as I know in Sensititre Yeast One (SYO) system the susceptibility to itraconazole is also investigated.

Lines 128-130  italic should be used for yeasts names

It is not clear for me why the authors decided to investigated the presented mechanisms of resistance – e.g. for C. glabrata PDR1 gene was sequenced. I also do not think that sequencing of CDR1 gene was a good idea – the resistance is mostly the consequence of overexpression of genes coding for drug transporters (including CDR1) not mutations within these genes (please see the articles of Maheronnaghsh M (2022); or  Masakazu Niimi (2004) – C. albicans and Gucwa and coworkers (2015) – for C. glabrata)

Results and discussion

Figure 1 presented in page 5 should by Figure 2. Would it be possible to prepare a larger version of this figure?

Line 171 “Two isolates of C. albicans were misidentified …” with which method and which of them gives “more important” result.

Figure 2 should be Figure 3 and Figure 3 should be Figure 4. In figures 1 and 2 (numbers presented in the manuscript) it should be clearly written what is presented on y axis.

Results for itraconazol are not presented in the tables.

As mentioned above. It is not clear for me why the authors decided to investigate these mechanisms of resistance.

Discussion – well prepared

Conclusions – if possible should be shortened.  

Final decision – minor revision  

Reviewer 2 Report

Article

Nationwide Surveillance of Antifungal Resistance of Candida 2 Bloodstream Isolates in South Korean Hospitals: Two Year Re- 3 port from Kor-GLASS

Dear Authors,

I consider the content of the paper pertinent and valuable, to have updated data about the epidemiology of hospital infections and, above all, the impact of candidemia in these scenarios.

In methodology, it is necessary to describe or cite the primers used for the PCR of the evaluated genes.

Include in supplementary material the sequences that were used for the identifications. I would like you to review again the one that you identify as C. haemouloni, since in its resistance profile it looks like a C. auris.

Similarly, in supplementary material, have available the sequences of the genes where the mutations are reported.

I also consider relevant the characterization of some genes associated with the described resistances. However, it is necessary to include this topic in the introduction, to know if the outcomes of the patients were related to the established therapies, and to include in the discussion some options in the case of resistant strains that are not supported by mutations.

For hospital H, it is in the only case in which C. albicans does not exceed Candida no albicans, something that they consider explaining this result at the epidemiological level will not be included, as well as the age range between 31-40 years where neither is C. albicans the majority.

And lastly, I don't know if the authors consider it appropriate to know if there is a common strategy for the establishment of antifungal therapy in Korea.

Reviewer 3 Report

Excellent examination of isolates and mechanisms of resistance.  HO and ICU parts insightful. 

1. Since so few pediatric isolates maybe helpful to readers not familiar to South Korea Health system to add one sentence in the discussion if children are cared for at other hospitals (Children's Hospital) or not. 

2.  Is information of whether patient was on antifungal prophylaxis at the time of the infection available in this data? If so, would it be additive to include?

3. Minor typo on page 2, Line 60: "associated' misspelled
